# PHYSICS AWARE NEURAL NETWORKS : DENOISING FOR MAGNETIC NAVIGATION

## ABSTRACT

Magnetic-anomaly navigation, which leverages small-scale local variations in the Earth's magnetic field, has emerged as a promising alternative for environments where GPS signals are unavailable or compromised. Airborne systems face a fundamental challenge in extracting the necessary geomagnetic field data: the aircraft itself induces magnetic noise. Although the classical Tolles-Lawson model addresses this, it inadequately handles stochastically corrupted magnetic data needed for operational navigation. To handle stochastic noise, we propose a novel approach using two physics-based constraints: divergence-free vector fields and E(3)-equivariance. Our constraints guarantee that the generated magnetic field obeys Maxwell's equations and ensure that output changes appropriately with sensor position/orientation. The divergence-free constraint is implemented by defining a neural network outputting vector potential $A$, with the magnetic field constructed as its curl. For E(3)-equivariance, we use tensor products of geometric tensors that are representable using spherical harmonics with well-known rotational transformations. By enforcing physical consistency and constraining the space of admissible field functions, our formulation acts as an implicit regularizer, improving spatio-temporal performance. We conduct ablation studies evaluating these constraints' individual and combined effects across CNNs, MLPs, Liquid Time Constant Models, and Contiformers. We note that continuous-time dynamics and long-term memory are critical for modelling magnetic time-series data; the Contiformer architecture, which inherently possesses both, surpasses state-of-the-art methods in our experiments. To handle data scarcity, we develop synthetic datasets by utilising the World Magnetic Model (WMM) in conjunction with time-series conditional GANs, generating realistic and temporally consistent magnetic field sequences spanning various trajectory patterns and environmental scenarios. Our experiments demonstrate that embedding these constraints significantly improves predictive accuracy and physical plausibility, outperforming the state-of-the-art across both classical and unconstrained deep learning approaches. Our code is available at `https://github.com/darksideofmoon123456/summerhouse`.

## 1 INTRODUCTION

Satellite based navigation systems, commonly referred to as Global Navigation Satellite System (GNSS), is now ubiquitous, underpinning activities from commerce to defense. However, today there is an increased threat of jamming and spoofing of GNSS signals, affecting civilian and military aircraft alike. Between Aug 21 - June 24, over 580,000 GNSS signal loss events on aircraft were reported IATA (2024). Fallback systems such as inertial sensors are limited due to sensor drift which increases over time Woodman (2007) Groves (2015).

A promising alternative is *magnetic-anomaly navigation*, which utilizes small-scale local anomalies in Earth's magnetic field (which have well-defined geographical signatures) to overcome the limitations of traditional navigation systems Canciani & Raquet (2020). Simply put, with a known background magnetic field, sufficiently precise local measurements can be used for geolocation and navigation. However, magnetic-anomaly navigation is limited because the magnetic field sensors onboard the aircraft measure magnetic fields produced by the aircraft's chassis, engines and instruments, which are essentially noise in the measurement Gnadt (2022a) Nerrise et al. (2023).

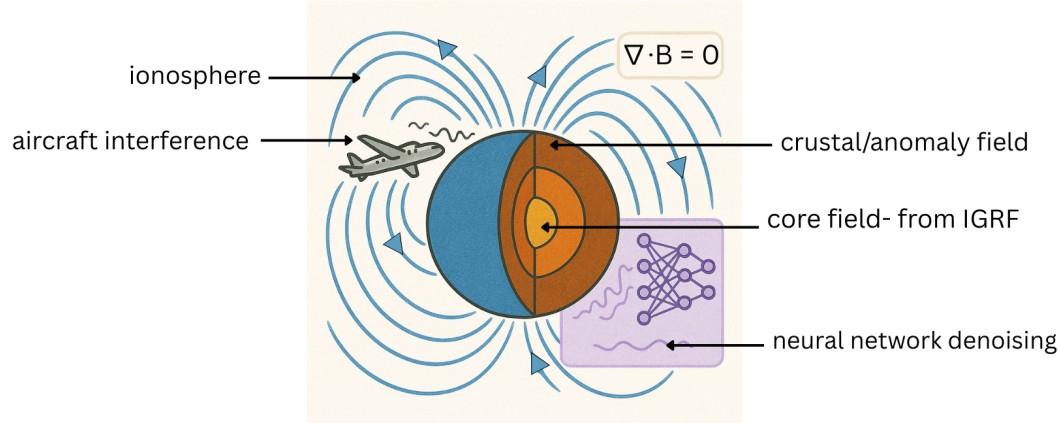

Figure 1: An illustrative diagram of our problem setting.

This limitation also applies to platforms other than aircraft, such as ships, unstaffed aerial vehicles (UAVs), etc. Extracting the real geomagnetic field data (i.e., the field that would have been measured if the aircraft did not exist) from onboard sensor measurements containing these superimposed magnetic components is difficult, especially for stochastic noise. Without denoising, navigation becomes error-prone and impractical.

Prior work in the area (detailed in the next section) has struggled to handle such noise, even with deep-learning based approaches. We aim to approach the problem from its fundamentals: Given the nature of the physical phenomenon we are measuring, what properties should an ideal network have to solve it?

Contributions

1. We introduce two physics-aware constraints for magnetic navigation: (i) divergence-free vector fields and (ii) E(3)-equivariance. These ensure predictions obey Maxwell's second law of electromagnetism ($\nabla \cdot \mathbf{B} = 0$) and respond correctly to changes in position and orientation. By enforcing physical consistency, we implicitly regularize the model, improving spatiotemporal performance.

2. We identify key properties for magnetic navigation—continuous-time dynamics and long-term memory suitable for spatiotemporal settings—and find appropriate architectures. Our ablation studies show how each constraint improves performance alone and in combination.

3. To address data scarcity, we generate synthetic magnetic field sequences using the World Magnetic Model and conditional GANs, covering diverse trajectories and environments.

4. We design, implement, and evaluate a novel system for denoising magnetic field sensor data for navigation, outperforming the state-of-the-art across both classical and unconstrained deep learning approaches.

## 2 RELATED WORK AND PRELIMINARIES

### 2.1 FILTERING BASED APPROACHES

Principal Component Analysis (PCA) decomposes flight lines or spatio-temporal data into principal components, isolating dominant noise trends - e.g., slowly varying diurnal baselines. Qiao et al. (2022) proposed an RPCA-based aeromagnetic compensation method, which improved outlier robustness and accuracy over traditional models. Simple moving-average filters better adapt to smooth, time-varying changes. Several finite impulse response (FIR) moving-average filters and an exponential moving average (IIR) for smoothing magnetometer signals in an unmanned ground vehicle were compared Pereira et al. (2024). Hybrid approaches such as wavelet–Wiener algorithms

also improve noise reduction, while adaptive Kalman filters can struggle with nonlinear coupling of aircraft interference Canciani (2022).

## 2.2 TOLLES-LAWSON MODEL

The linear aeromagnetic compensation model for scalar magnetometers was introduced by Tolles & Lawson (1950), with patents for associated hardware Tolles (1954; 1955). Enhancements using sinusoidal flight maneuvers improved observability of model coefficients Leliak (1961). Various extensions refined the model, but stochastic noise remains inadequately handled Leach (1979); Gu et al. (2013); Webb & White (2014); Wu et al. (2018); Gnadt et al. (2022). Shallow, feature-informed networks can improve performance while maintaining interpretability Gnadt (2022b).

## 2.3 DEEP-LEARNING BASED APPROACHES

Introduced neural networks for compensating aircraft magnetic interference, estimating the complete magnetic field as a combination of crustal, diurnal, and platform-generated components. Modern deep learning methods, including DeepMAD Xu et al. (2020), CNNs, recurrent networks, and Liquid Time-Constant (LTC) networks, capture nonlinear interference patterns more effectively than classical methods. Recently, there has been concurrent work on using similar physics-aware approaches to constraining for finding non-singularities in Navier-Stokes equations. Wang et al. (2025)

# 3 OUR APPROACH

We present a first-principles approach to magnetic field denoising that enforces physical laws and handles real-world magnetometer noise. First, we integrate physics-aware constraints directly into neural networks, for extraction of geomagnetic signals while preserving physical consistency. Second, we identify two key physical properties of magnetic fields essential for effective network design.

Magnetic fields in free space must satisfy (1) Maxwell's equations, specifically: divergence-free constraint ($\nabla \cdot \mathbf{B} = 0$), which is depicted in Figure 2 and (2) E(3)-equivariance under rigid transformations. By embedding these constraints directly into the network architecture rather than as soft penalties, we constrain the space of admissible field functions to physically plausible solutions. One of the fundamental properties of magnetic data is that they are irregularly sampled, with measurement times $t_i$ separated by $\Delta t_i = t_{i+1} - t_i$. Therefore, to correctly handle data, we model the hidden state $z(t)$ via a neural ODE $\dot{z}(t) = f_\theta(z(t), t)$ which naturally evolves in continuous time across each arbitrary interval $\Delta t_i$. Magnetometer noise exhibits long-term dependencies, as its autocorrelation $R(\tau) = \mathbb{E}[n(t)\,n(t+\tau)]$ decays slowly (e.g., $R(\tau) \sim e^{-\lambda|\tau|}$ for small $\lambda$), necessitating long-term memory for modeling and denoising scenarios. Our entire denoising pipeline is given in Figure 4.

## 3.1 DIVERGENCE-FREE CONSTRAINT

We parametrize $\mathbf{B}$ via a vector potential $\mathbf{A}$ such that $\mathbf{B} = \nabla \times \mathbf{A}$. We do this by defining a neural network $\mathcal{A}_\theta : \mathbb{R}^3 \to \mathbb{R}^3$ that outputs the vector potential $\mathbf{A}_\theta(\mathbf{x})$. The magnetic field is then constructed as the curl of this potential:

$$\mathbf{B}_\theta(\mathbf{x}) := \nabla \times \mathbf{A}_\theta(\mathbf{x}) \tag{1}$$

This guarantees that $\nabla \cdot \mathbf{B}_\theta = 0$ identically, due to the vector calculus identity $\nabla \cdot (\nabla \times \mathbf{A}) = 0$ for all smooth vector fields $\mathbf{A}$. In 3D Euclidean space $\mathbb{R}^3$, the magnetic field can be treated as a differential 2-form $\mathbf{B} \in \Omega^2(\mathbb{R}^3)$, and the vector potential as a 1-form $\mathbf{A} \in \Omega^1(\mathbb{R}^3)$. The curl operation corresponds to applying the exterior derivative $d$ followed by the Hodge star operator $\star$:

$$\mathbf{B} = \star d\mathbf{A} \tag{2}$$

This geometric formulation further shows why $\nabla \cdot \mathbf{B} = 0$. In differential forms, the divergence of a vector field (as a 1-form) is:

$$\nabla \cdot \mathbf{B} = \star d \star \mathbf{B} \tag{3}$$

Since $\mathbf{B} = \star d\mathbf{A}$, we have:

$$\nabla \cdot \mathbf{B} = \star d \star (\star d\mathbf{A}) = \star dd\mathbf{A} = 0 \tag{4}$$

The last equality holds because $d^2 = 0$. This shows that our approach is compatible with differential forms used in recent works on neural PDEs and conservation laws Richter-Powell et al. (2022), while allowing direct incorporation of physical priors into model design.

The network $\mathcal{A}_\theta$ is implemented using a standard feedforward or equivariant architecture. During inference and training, $\mathbf{B}_\theta$ is computed using automatic differentiation libraries to calculate the spatial derivatives needed for the curl:

$$[\nabla \times \mathbf{A}_\theta]_i = \epsilon_{ijk} \frac{\partial \mathcal{A}_\theta^k}{\partial x^j}, \tag{5}$$

where $\epsilon_{ijk}$ is the Levi-Civita symbol, and Einstein summation is implied. The vector potential representation introduces gauge freedom: for any scalar function $\phi$, both $\mathbf{A}$ and $\mathbf{A} + \nabla\phi$ yield the same magnetic field. This non-uniqueness might seem problematic, but it provides additional flexibility for optimization. Standard gradient descent naturally selects smooth solutions within the gauge equivalence class.

**Total magnetic field passing through a given area or volume**

Magnetic flux forms continuous closed loops.
Flowing from the north pole to the south pole externally.
Returning through the magnet internally.

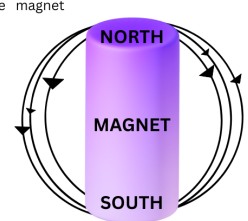

The north and south poles of a magnet are inseparable,
and thus no isolated magnetic charge can occur within the field.

Figure 2: Maxwell's 2nd Law of Electromagnetism. Magnetic field lines have no beginning or end, but exist as loops or extend to infinity: magnetic monopoles cannot exist.

## 3.2 E(3)-EQUIVARIANCE

Vehicle-mounted magnetometers measure fields in their local reference frame, which changes with orientation. So, a physically consistent model must transform under rotations and translations of the sensor platform. Formally, if $g = (R, \mathbf{t}) \in$ E(3) represents a rigid transformation, then:

$$\mathbf{B}'(R \cdot \mathbf{r} + \mathbf{t}) = R \cdot \mathbf{B}(\mathbf{r}) \tag{6}$$

Or, for a neural network: Let $G$ be a group with group elements $g \in G$. For transformations $\Psi_g : X \to X$ on input domain $X$, a mapping $\varphi : X \to Y$ satisfies $G$-equivariance when there exist transformations $\Phi_g : Y \to Y$ on output domain $Y$ such that Satorras et al. (2021):

$$\varphi(\Psi_g(x)) = \Phi_g(\varphi(x)) \tag{7}$$

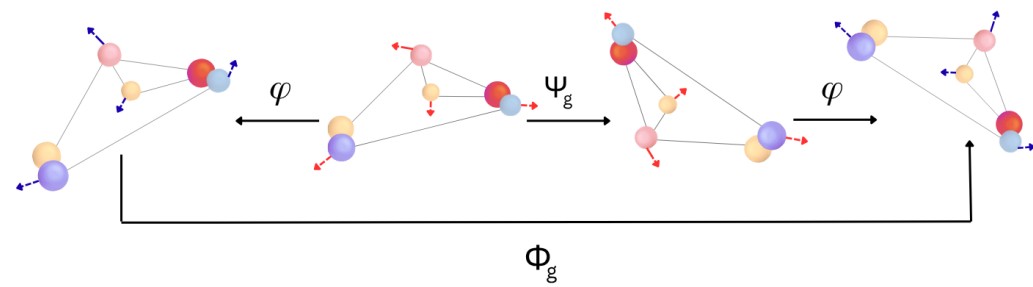

Figure 3: E(3)-equivariance demonstration: applying geometric transformations before or after the neural network $\varphi$ yields equivalent results, ensuring consistent magnetic field predictions regardless of coordinate system orientation.

(See Figure 3). This ensures that our predictions remain consistent regardless of sensor pose, since orientation varies continuously during navigation. We implement E(3)-equivariance using geometric tensors, which can be represented using spherical harmonics:

$$\mathbf{T}^{(\ell)}(\mathbf{r}) = T(r)\mathbf{Y}^{(\ell)}(\hat{\mathbf{r}}) \tag{8}$$

where $\mathbf{Y}^{(\ell)} = (Y_\ell^{-\ell}, Y_\ell^{-\ell+1}, \ldots, Y_\ell^\ell)^T$ and $T(r)$ is a radial function. This works because spherical harmonics are precisely the basis functions for irreducible representations (irreps) of $SO(3)$. A standard layer is given by: $\mathbf{y} = \sigma(W\mathbf{x} + \mathbf{b})$ – but this is not equivariant! If we rotate the input, the output does not rotate consistently. We can create a general E(3)-equivariant layer in a neural network through the following linear operation between geometric tensors:

$$\mathbf{T}_{\text{out}}^{(\ell_{\text{out}})} = \sum_{\ell_{\text{in}}} \sum_\ell W^{(\ell_{\text{out}},\ell_{\text{in}},\ell)} [\mathbf{T}_{\text{in}}^{(\ell_{\text{in}})} \otimes \mathbf{Y}^{(\ell)}]^{(\ell_{\text{out}})} \tag{9}$$

**Input**: $\mathbf{T}_{\text{in}}^{(\ell_{\text{in}})}$ is a geometric tensor of type $(\ell_{\text{in}})$
**Spherical Harmonics**: $\mathbf{Y}^{(\ell)}$ provides the geometric information about relative positions
**Tensor Product**: $[\mathbf{T}_{\text{in}}^{(\ell_{\text{in}})} \otimes \mathbf{Y}^{(\ell)}]^{(\ell_{\text{out}})}$ combines them using Clebsch-Gordan coefficients
**Weights**: $W^{(\ell_{\text{out}},\ell_{\text{in}},\ell)}$ are scalar weights (learnable parameters)

The tensor product $[\mathbf{T}^{(\ell_1)} \otimes \mathbf{Y}^{(\ell_2)}]^{(\ell)}$ is computed as:

$$[\mathbf{T}^{(\ell_1)} \otimes \mathbf{Y}^{(\ell_2)}]_m^{(\ell)} = \sum_{m_1=-\ell_1}^{\ell_1} \sum_{m_2=-\ell_2}^{\ell_2} \langle \ell_1 m_1 \ell_2 m_2 | \ell m \rangle T_{m_1}^{(\ell_1)} Y_{\ell_2}^{m_2} \tag{10}$$

For a relative position vector $\mathbf{r}_{ij} = \mathbf{r}_j - \mathbf{r}_i$ we have $Y_\ell^m(\mathbf{r}_{ij}) = Y_\ell^m(\theta_{ij}, \phi_{ij})$, where $(\theta_{ij}, \phi_{ij})$ are the spherical angles of $\hat{\mathbf{r}}_{ij} = \mathbf{r}_{ij}/|\mathbf{r}_{ij}|$. For a node $i$ with neighbors $\mathcal{N}(i)$:

$$\mathbf{T}_i^{(\ell_{\text{out}})} = \sum_{j \in \mathcal{N}(i)} \sum_{\ell_{\text{in}}} \sum_\ell W^{(\ell_{\text{out}},\ell_{\text{in}},\ell)} \times [\mathbf{T}_j^{(\ell_{\text{in}})} \otimes \mathbf{Y}^{(\ell)}(\hat{\mathbf{r}}_{ij})]^{(\ell_{\text{out}})} \tag{11}$$

See Appendix A for proof of E(3)-equivariance. We modify this for the LTC and Contiformer architectures, as they operate on continuous-time dynamics governed by Neural ODEs. For the Contiformer, we embed the tensor product in the attention integral:

$$\alpha(\mathbf{r}, t; \mathbf{r}_i, t_i) = \frac{1}{t - t_i} \int_{t_i}^t \sum_{\ell_q, \ell_k, \ell} \mathbf{Q}^{(\ell_q)}(\mathbf{r}, \tau)$$

$$\cdot \left[\mathbf{K}^{(\ell_k)}(\mathbf{r}_i, \tau) \otimes \mathbf{Y}^{(\ell)}(\widehat{\mathbf{r} - \mathbf{r}_i})\right]^{(\ell_q)} d\tau \tag{12}$$

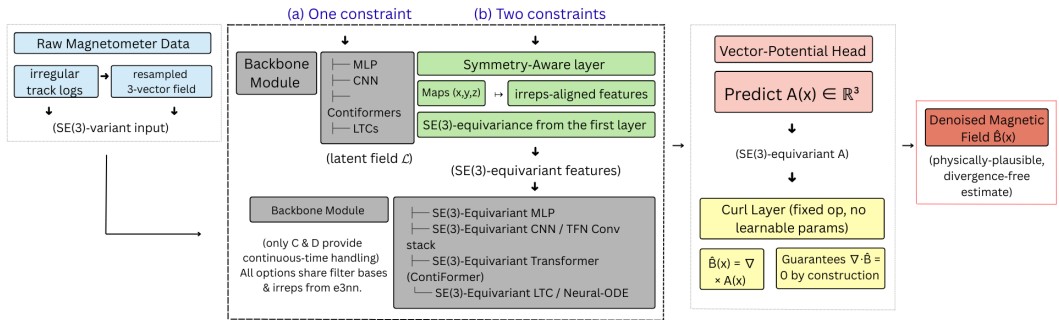

Figure 4: Denoising Pipeline with Divergence-Free and E(3)-Equivariant Constraints

and obtain the final output:

$$\mathbf{T}_{\text{out}}^{(\ell_{\text{out}})}(\mathbf{r}, t) = \sum_{i=1}^{N} \sum_{\ell_v, \ell_{\text{mix}}} W_{\text{out}}^{(\ell_{\text{out}}, \ell_v, \ell_{\text{mix}})}$$

$$\times \left[ \alpha(\mathbf{r}, t; \mathbf{r}_i, t_i) \mathbf{V}_{\exp}^{(\ell_v)}(\mathbf{r}, t; \mathbf{r}_i, t_i) \otimes \mathbf{Y}^{(\ell_{\text{mix}})}(\widehat{\mathbf{r} - \mathbf{r}_i}) \right]^{(\ell_{\text{out}})} \tag{13}$$

For LTC, each hidden unit $i$ is governed by the ODE Hasani et al. (2022):

$$\tau_i(x_t, h_t)\dot{h}_i(t) = -h_i(t) + \sigma\left(W_{x,i}x_t + W_{h,i}h_t + b_i\right) \tag{14}$$

Equation 14 is linear in $h_i$, which means it has a closed-form solution over an event span $\Delta t$:

$$h_i(t + \Delta t) = e^{-\Delta t/\tau_i} h_i(t) + \left(1 - e^{-\Delta t/\tau_i}\right) u_i(t),$$
$$u_i(t) = \sigma(\cdots) \tag{15}$$

To achieve E(3)-equivariance in the LTC model, we treat the state vector $h_t$ as a collection of geometric tensors: 64 copies of scalar irreps (0e) and vector irreps (1o) e3nn Developers (2025) corresponding to the magnetic field components. Since scalars are rotationally invariant and vectors transform as $\ell = 1$ geometric tensors, the ODE dynamics in Equation 15 automatically preserve E(3)-equivariance by evolving each irreducible representation independently. This approach never mixes channels of different tensor types (0e with 1o), maintaining the geometric structure throughout.

### 3.3 CONTINUOUS-TIME HANDLING

Magnetometer data is irregularly sampled, with measurement times $t_1$, $t_2$, ..., $t_N$ and unpredictable intervals $\Delta t_i = t_{i+1} - t_i$ arising from sensor jitter, asynchronous system clocks, and platform motion. This irregular sampling violates the fixed-step assumption. Additionally, real-world magnetic measurements are corrupted by structured, time-varying noise: e.g., eddy-current noise arises from induced currents in nearby conductors and obeys diffusion-type ODEs, leading to non-stationary distortions; temperature-driven bias and external electromagnetic interference introduce further drift and slowly varying biases that cannot be modelled as white noise. To address irregular sampling and ODE-governed noise, we model the latent state $z(t) \in \mathbb{R}^d$ via a continuous-time neural ODE, $\frac{dz}{dt} = f_\theta(z(t), t)$, where $f_\theta$ is a neural network. At each $t_i$, the model maps $z(t_i)$ to observation space as $\hat{x}(t_i) = g_\phi(z(t_i))$ and penalizes the discrepancy $\|\hat{x}(t_i) - x_{\text{meas}}(t_i)\|$ in the loss.

### 3.4 LONG-TERM MEMORY

Magnetic noise sources like eddy currents, sensor drift, and thermal bias often have long-lasting effects.

Table 1: CNN and MLP results.

| | | CNN | | | | MLP | | | |
| --- | --- | --- | --- | --- | --- | --- | --- | --- | --- |
| | | Base | Divergence-Free | E(3)-Equivariance | Both | Base | Divergence-Free | E(3)-Equivariance | Both |
| RMSE (nT) | Training | 55.36 | 36.58 | 33.27 | 30.19 | 65.23 | 40.12 | 37.45 | 25.67 |
| | Testing | 60.29 | 41.14 | 38.61 | 35.48 | 70.47 | 45.28 | 42.19 | 30.34 |
| SNR (dB) | Training | 45.18 | 48.78 | 49.60 | 50.45 | 43.75 | 47.98 | 48.57 | 51.85 |
| | Testing | 44.44 | 47.76 | 48.31 | 49.04 | 43.08 | 46.93 | 47.54 | 50.40 |

Table 2: LTC and Contiformer results. RMSE ranges from 13.09 to 42.31 nT and SNR ranges from 47.51 to 57.70 dB.

| | | LTC | | | | Contiformer | | | |
| --- | --- | --- | --- | --- | --- | --- | --- | --- | --- |
| | | Base | Divergence-Free | E(3)-Equivariance | Both | Base | Divergence-Free | E(3)-Equivariance | Both |
| RMSE (nT) | Training | 38.74 | 30.85 | 28.43 | 24.92 | 19.04 | 16.16 | 15.22 | 13.09 |
| | Testing | 42.31 | 34.67 | 32.16 | 28.55 | 21.57 | 18.62 | 17.75 | 15.09 |
| SNR (dB) | Training | 48.28 | 50.26 | 50.97 | 52.11 | 54.45 | 55.87 | 56.40 | 57.70 |
| | Testing | 47.51 | 49.24 | 49.90 | 50.93 | 53.37 | 54.64 | 55.06 | 56.47 |

For example, eddy currents can last minutes, while temperature bias evolves over hours. These slow dynamics cause long-range autocorrelation, with noise at time $t$ still correlated with much earlier noise.

Sensor offsets and low-frequency interference accumulate rather than reset. Thus, we must maintain a latent state $z(t)$ capable of storing long-term context, so past noise remains accessible for accurate prediction.

### 3.5 Synthetic Data Generation using Conditional GAN

MagNav datasets are limited, with only a handful of publicly available flight recordings. The data comprises multichannel magnetometer sequences recorded at $10\,\text{Hz}$ aboard fixed-wing aircraft; the limited available datasets constrain supervised learning models for magnetic denoising. To expand this corpus, we train a conditional Generative Adversarial Network (cGAN) that learns $p(\mathbf{x} \mid c)$, the joint distribution of windowed sensor readings $\mathbf{x} \in \mathbb{R}^{T \times C}$ conditioned on a discrete context label $c$ (e.g. flight ID, attitude bin). The generator $G_\theta$ accepts latent noise $\mathbf{z} \sim \mathcal{N}(\mathbf{0}, I)$ and an embedding $\mathbf{e}_c$ and produces synthetic sequences $\tilde{\mathbf{x}} = G_\theta(\mathbf{z}, \mathbf{e}_c)$; the discriminator $D_\phi$ receives $(\mathbf{x}, \mathbf{e}_c)$ and outputs two heads: $D_{\text{adv}} \in [0,1]$ for real/fake and $D_{\text{cls}} \in \Delta^{K-1}$ for class prediction. The GAN design is as follows: Generator: 2-layer LSTM ($h = 256$) with linear expansion of $[\mathbf{z}; \mathbf{e}_c]$ to $T$ time steps, output projected to $C = 4$ channels and bounded by $\tanh$. Discriminator: matching LSTM depth, spectral normalization on recurrent weights, final hidden state feeds dual heads.

Furthermore, training minimizes the discriminator loss

$$\mathcal{L}_D = -\mathbb{E}_{(\mathbf{x},c)} \log D_{\text{adv}} + \mathbb{E}_{(\mathbf{z},c)} \log(1 - D_{\text{adv}}) + \lambda \mathcal{L}_{\text{cls}}^D \tag{16}$$

and generator loss

$$\mathcal{L}_G = -\mathbb{E}_{(\mathbf{z},c)} \log D_{\text{adv}} + \lambda \mathcal{L}_{\text{cls}}^G, \tag{17}$$

where $D_{\text{adv}}$ denotes the adversarial head output, $\mathcal{L}_{\text{cls}}^D$ and $\mathcal{L}_{\text{cls}}^G$ are cross-entropy classification losses on real and synthetic samples, respectively, and $\lambda = 1$. We employ AdamW optimizer ($\eta = 10^{-4}$, $\beta_{1,2} = (0.5, 0.999)$) with gradient clipping at $1.0$ and one-sided label smoothing of $0.9$.

## 4 Experiments

We evaluate our denoising system with three goals: (i) Noise removal, (ii) Preservation of physical properties – divergence-free structure and E(3) equivariance, and (iii) Verifying that models with long-term memory and continuous-time handling outperform those without.

### 4.0.1 Data Pre-Processing

While raw magnetometer data can be used directly, we implemented physics-based corrections to remove known interference. Transient Level compensation eliminated aircraft-induced noise. IGRF

correction removed temporal variations from the Earth's core field, isolating crustal signatures. Diurnal corrections minimized time-dependent fluctuations and measurement noise.

### 4.0.2 TRAINING/TESTING

We tested 16 models, combining four backbone architectures (MLP, 1D CNN, LTC, and Contiformer) with 4 layers of constraint (Plain, Div-Free, E3NN, and E3NN + Div-Free) applied orthogonally. Each model was trained on an NVIDIA RTX A6000 GPU using consistent parameters, including 10-second windows, batch size of 128, 40 epochs, Adam optimizer with exponential learning rate decay, and early stopping based on validation RMSE.

### 4.0.3 BASELINES AND METRICS

A significant bottleneck in evaluating magnetic denoising and compensation techniques is due to the lack of standardized datasets and metrics. Image-based methods often use PSNR, SSIM, and RMSE, while sensor-based approaches favor STD, NRF, and IR Pereira et al. (2024); Wang et al. (2022). Datasets vary from images, to time-series, to UAV surveys. For our MagNav flight data and similar synthetic datasets, traditional metrics like SPE, PSNR, and SSIM are ill-suited due to the non-stationary, time-series nature. Instead, we use time-domain metrics: Root Mean Squared Error (RMSE) to measure reconstruction accuracy, and Signal-to-Noise Ratio (SNR) to evaluate denoising quality.

$$\text{RMSE} = \sqrt{\frac{1}{T} \sum_{t=1}^{T} \left\| \hat{\mathbf{B}}(t) - \mathbf{B}(t) \right\|_2^2}$$

while SNR measures relative noise suppression: Higher SNR values indicate better performance - for eg., an SNR of 100 dB means our signal is $10^{10}$ times stronger than the error, while 80 dB means it's $10^8$ times stronger, so even a 20 dB difference represents a 100-fold improvement in signal quality.

$$\text{SNR} = 10 \cdot \log_{10} \left( \frac{\sum_{t=1}^{T} \|\mathbf{B}(t)\|_2^2}{\sum_{t=1}^{T} \left\| \hat{\mathbf{B}}(t) - \mathbf{B}(t) \right\|_2^2} \right)$$

$\mathbf{B}(t)$ = true field; $\hat{\mathbf{B}}(t)$ = field generated by the NN.

## 5 RESULTS

We evaluated four architectures: MLP, CNN, LTC, and ContiFormer on the Flt1005, Flt1006, Flt1007, 2017, 2015, 2016 from MagNav and FltS005, FltS006, FltS007 from the conditional GAN synthetic dataset. Without constraints, performance varied widely: MLP performed worst (Train RMSE: 65.23; Test: 70.47), while ContiFormer was best (Train: 25.38; Test: 28.76). CNN and LTC achieved intermediate results (Test RMSEs: 60.29 and 42.31), consistent with prior work Nerrise et al. (2024). (See Table.) Adding physics-aware constraints consistently improved performance. The combination of divergence-free and equivariance constraints yielded the largest gains: MLP test RMSE dropped by 57%, and ContiFormer improved by 30%. These results highlight the benefit of embedding physical priors, especially in data-scarce or noisy settings.

## 6 DISCUSSION

Our experiments show that neural networks with divergence-free and equivariance constraints consistently outperform unconstrained architectures for magnetic field estimation. Though the reasons are more nuanced than they first appear. While magnetic fields inherently obey these constraints, real-world measurements are corrupted by sensor noise, platform interference, and calibration errors, which introduce physically invalid divergences. Magnetic fields can be decomposed into solenoidal, gradient, and harmonic components via the Helmholtz-Hodge decomposition Bhatia et al. (2013).

*True magnetic fields lie entirely in the solenoidal subspace* due to Gauss's law, while the gradient component captures noise that violates this constraint. Thus, forcing network outputs to be divergence-free effectively removes this non-physical gradient noise. The $E(3)$-equivariance constraint complements this by averaging over rotations, suppressing noise components that do not transform correctly under coordinate changes. Since most disturbances arise from hardware orientation, not the field itself, this symmetry-based filtering enhances robustness while preserving the true rotational structure of the magnetic field.

## 7 CONCLUSION

We address the challenge of magnetic anomaly navigation, highlighting the importance of denoising magnetometer data for downstream performance. We develop neural architectures tailored to the problem's structure-marked by continuous-time dynamics and long-range dependencies—and introduce two physics-aware constraints: divergence-freedom and $E(3)$-equivariance. Our experiments show that combining these priors with suitable architectures yields significant gains over the state-of-the-art. This line of research demonstrates that physics constraining can be effectively applied for other purposes as well, utilizing different architectures for various spatio-temporal systems beyond magnetic anomaly navigation. The incorporation of fundamental physical and mathematical principles into network architectures provides a generalizable framework for diverse dynamical systems.

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

## A APPENDIX

We claim that the expression given in equation 11 is E(3)-equivariant. Let us formally prove this.

Consider a transformation $g = (R, \mathbf{t}) \in E(3)$.

1. Under the transformation:
   - Positions: $\mathbf{r}_i' = R\mathbf{r}_i + \mathbf{t}$
   - Relative positions: $\mathbf{r}_{ij}' = \mathbf{r}_j' - \mathbf{r}_i' = R(\mathbf{r}_j - \mathbf{r}_i) = R\mathbf{r}_{ij}$
   - Unit vectors: $\hat{\mathbf{r}}_{ij}' = R\hat{\mathbf{r}}_{ij}$

2. Spherical harmonics transform as:

$$\mathbf{Y}^{(\ell)}(\hat{\mathbf{r}}_{ij}') = \mathbf{Y}^{(\ell)}(R\hat{\mathbf{r}}_{ij}) = D^{(\ell)}(R)\mathbf{Y}^{(\ell)}(\hat{\mathbf{r}}_{ij}) \tag{18}$$

3. Input tensors transform as:

$$\mathbf{T}_j^{\prime(\ell_{\text{in}})} = D^{(\ell_{\text{in}})}(R)\mathbf{T}_j^{(\ell_{\text{in}})} \tag{19}$$

4. The tensor product preserves equivariance:

$$\begin{aligned}
[\mathbf{T}_j^{\prime(\ell_{\text{in}})} \otimes \mathbf{Y}^{(\ell)}(\hat{\mathbf{r}}_{ij}')]^{(\ell_{\text{out}})} &= D^{(\ell_{\text{out}})}(R) \\
&\times [\mathbf{T}_j^{(\ell_{\text{in}})} \otimes \mathbf{Y}^{(\ell)}(\hat{\mathbf{r}}_{ij})]^{(\ell_{\text{out}})}
\end{aligned} \tag{20}$$

5. Since weights are scalars, the final result is:

$$\mathbf{T}_i^{\prime(\ell_{\text{out}})} = D^{(\ell_{\text{out}})}(R)\mathbf{T}_i^{(\ell_{\text{out}})} \tag{21}$$

This proves E(3)-equivariance.

