# OpenReview forum: "Physics Aware Neural Networks : Denoising for Magnetic Navigation"
_ICLR.cc/2026/Conference — Submitted to ICLR 2026_

### Official Review · Reviewer_pXsP · 2025-10-25

**Soundness:** 3
**Presentation:** 1
**Contribution:** 2
**Rating:** 2
**Confidence:** 3

**Summary:**

The manuscript presents a method for enforcing hard divergence-free and E(3)-equivariant constraints in neural network outputs for magnetic navigation denoising. The authors evaluate the effectiveness of these constraints across four different network architectures using the MagNav and GAN-synthetic datasets. The reported results demonstrate the benefit of incorporating the proposed physics-based constraints.

**Strengths:**

1. The technical implementation of the divergence-free and E(3)-equivariant constraints is clearly described, supported by sufficient mathematical detail.
2. The introduction effectively outlines the challenges in magnetic navigation. Even for readers without prior expertise in this domain, the problem setup and motivation are easy to follow.

**Weaknesses:**

1. The primary weakness of the manuscript lies in its organization and lack of clarity regarding how the proposed method performs denoising. For readers unfamiliar with magnetic navigation, it is difficult to understand the actual application workflow. For example, after Sections 3.1 and 3.2, it is clear that the network takes the spatial location $x$ as input and outputs a divergence-free, E(3)-equivariant magnetic field. However, it remains unclear how this network is used for denoising: What are the input and target data? What is the loss function? How does the network remove noise from the data in practice?
2. The literature review and related work sections are underdeveloped. The ideas of imposing hard physical constraints and enforcing E(3)-equivariance are not novel, and relevant prior work is insufficiently discussed. This omission weakens the paper’s contribution and contextual grounding. For instance, similar divergence-free constraints have long been employed in fluid dynamics, such as predicting vorticity fields and reconstructing divergence-free velocity fields via the curl operation. The current work appears conceptually similar to these established approaches.

**Questions:**

1. What are the specific input and target datasets used during denoising training, and what loss function is optimized?
2. In line 171, the authors state: “Standard gradient descent naturally selects smooth solutions within the gauge equivalence class.” What does “smooth solutions” mean in this context? In a standard regression setup, the constant $A$ should be determined by the training data. How is this related to smoothness?
3. What is the latent state $z(t)$ mentioned in line 310? Which network generates this latent representation, and is it the same as the “Latent Field $\mathcal{L}$ in Figure 4?
4. The MagNav dataset appears to be missing a proper citation. Please include a reference to its source.

---

### Official Review · Reviewer_k9Af · 2025-10-31

**Soundness:** 2
**Presentation:** 1
**Contribution:** 2
**Rating:** 2
**Confidence:** 3

**Summary:**

Authors propose, train and evaluate several architectures suitable for magnetic-anomaly navigation problem: magnetometer record magnetic field, neural network improve the recording by removing noise, the later data is used for navigation.

**Strengths:**

Authors study a reasonable applied problem with a lot of potential for improvement with deep learning techniques. Proposed methods - neural ODE to accommodate uneven sampling rate, transformer for long term memory, architectures with $E(3)$-equivariance to correctly handle the change in the orientation of the measuring device - all appear to be reasonable.

**Weaknesses:**

I see two major problems:
1. Authors failed to explain how their architectures are working: what is the input, what is the output, what are the layers, etc.
2. The data used by authors is synthetic, with no entirely clear explanation how this data was generated. Synthetic data also does not allow us to draw definitive conclusions about the practicality of proposed approaches.

**Questions:**

1. Lines 118-119, "Recently, there has been concurrent work on using similar physics-aware approaches to constraining for finding non-singularities in Navier-Stokes equations. Wang et al. (2025)". This is a suspicious reference:

   a. Referenced work is on the search for singularities in the Euler equation, not Navier-Stokes.

   b. It is on the use of physics-informed neural networks.

   c. The work is not related to magnetic inference, handling observation data, etc. It is an exploratory work for solving purely mathematical questions on the regularity of certain nonlinear PDEs.

2. Lines 148-162. Redundant explanation using Hodge star. Authors already explained how they generate divergent-free fields. Why introduce Hodge star? What is the purpose of this fragment?
3. Figure 3. $E(3)$ equivariance (looks more like $SO(3)$ equivariance) is illustrated using molecules. The main subject of the paper is denoising of magnetic field measurements. Why not illustrate equivariance with the magnetic field?
4. Equations (7) - (10), page 5. An explanation of the $E(3)$ equivariant layer is not clear. Suppose I have a measurement of the magnetic field $B\in\mathbb{R}^{3}$ at time $t$.

   a. How is this magnetic field processed by the equivariant layer?

   b. Where is the input, where is the output?

   c. What are $i$ and $j$ in $r_{ij}$? What are neighbours $N(i)$ of node $i$?

5. Appendix A. Authors reference Appendix A, but it is not available in the article. Supplementary materials are also absent.
6. Equations (12), (13).

   a. What are $Q^{(l_q)}$, $K^{(l_k)}$?

   b. What is attention integral?

   c. Where are observations of the magnetic field in the final output (13)?

7. Equation (14). What are $x_t$, $\tau_{i}(x_t, h_t)$?
8. Section 3.4. What kind of architecture is using this latent state? Is it LTS with latent state $h$? Is it MLP or CNN?
9. Section 3.5.

   a. Can the authors provide references to the dataset on magnetic navigation that they used?

   b. Authors mentioned that in those datasets observations are available at sampling rate $10Hz$. Given that data has regular sampling, why do authors assume irregular sampling rate in the construction of the architecture?

   c. What is class prediction in this context?

   d. Why is synthetic data useful for the task at hand? How can the generation of synthetic data capture noise of real devices?

10. Section 4.0.1. Can the author explain in detail how they preprocess data?

**Details Of Ethics Concerns:**

The research has potential for military use as authors themselves claim in the introduction:
"However, today there is an increased threat of jamming and spoofing of GNSS signals, affecting civilian and military aircraft alike."

---

### Official Review · Reviewer_Lzc1 · 2025-11-01

**Soundness:** 2
**Presentation:** 2
**Contribution:** 2
**Rating:** 2
**Confidence:** 5

**Summary:**

This paper addresses the challenge of denoising magnetometer data for magnetic anomaly navigation, a critical capability for environments where GPS signals are unavailable or compromised. The authors propose a novel approach that embeds two physics-based constraints directly into neural network architectures. The paper also emphasizes continuous-time dynamics and long-term memory for handling irregularly sampled magnetometer data. To address data scarcity, the authors develop synthetic datasets using the World Magnetic Model combined with time-series conditional GANs. Experimental results demonstrate that their physics-aware constraints improve predictive accuracy and physical plausibility across multiple architectures (CNNs, MLPs, LTCs, and Contiformers).

**Strengths:**

1. The paper addresses an important practical problem in navigation where GPS is unavailable or compromised, which has significant real-world applications in both civilian and military contexts.
2. The implementation of divergence-free fields through vector potential representation is mathematically sound and directly enforces a key physical constraint.

**Weaknesses:**

1. The paper combines several existing techniques (physics-informed neural networks, equivariant networks, continuous-time models). Thus, this paper has limited novelty and contribution.
2. The paper claims to "outperform the state-of-the-art" but does not compare against recent relevant methods that address similar problems. Therefore, the experiment is limited.
3. The paper mentions using "Flt1005, Flt1006" datasets but provides no information about their size, diversity, or how they were acquired.
4. The performance improvements could equally be attributed to the network architecture choices rather than the physics constraints themselves.
5. The authors should experiment on real datasets, not synthetic datasets.

**Questions:**

See Weaknesses.

---

### Official Review · Reviewer_wYsY · 2025-11-01

**Soundness:** 1
**Presentation:** 1
**Contribution:** 1
**Rating:** 0
**Confidence:** 4

**Summary:**

This work tackles magnetic-anomaly navigation for aircraft in GPS-denied settings, where local magnetic field variations can be used for positioning but aircraft-induced magnetic noise makes sensing difficult. Classical models like Tolles-Lawson aren’t robust to stochastic noise, so authors introduce a physics-informed neural approach that enforces two constraints: (1) the magnetic field must be divergence-free, implemented by predicting a vector potential whose curl gives the field, and (2) the model must be E(3)-equivariant, so outputs transform correctly under rotations and translations. These constraints act as an implicit regularizer, improving spatial and temporal generalization. Authors further evaluate multiple architectures (CNNs, MLPs, Liquid Time Constant Models, and especially Contiformers, which handle continuous-time dynamics and long-term memory) and find Contiformers outperform state of the art.

**Strengths:**

The paper does not demonstrate any clear strength.

**Weaknesses:**

1) The novelty claims are overstated. Enforcing divergence-free structure, embedding Euclidean symmetry, and adding physics constraints to neural predictors are all well-established ideas in the literature (e.g. PINNs, equivariant GNNs, divergence-free parameterizations in incompressible flow). The paper frames these as firsts in this domain and as magnetism-specific contributions, but the constraints they use are generic and not unique to magnetic navigation. The submission does not clearly articulate what is actually new beyond recombining known ingredients. If the contribution is something else it's not cleat at all

2) The paper’s empirical claims are not verifiable because the dataset is not described in enough detail to reproduce any experiment. The authors mention using several “publicly available flight recordings,” but they never provide any information regarding them.

3) The paper claims state-of-the-art performance but does not actually compare against the state of the art. Tables 1-2 only compare different variants of the authors’ own models. There are no numbers for standard magnetic compensation techniques.

4) The related work section does not situate this submission in the actual problem space. The paper spends space on generic physics-informed ML and equivariant networks, but it does not engage with recent work, as a result the paper does not provide any unsolved gaps in this specific area.

5) Almost every part is very poorly structured, from the abstract to figures.

We advise authors to read the following (these are the well established methods, pleas follow the citings to the recent papers and the current SOTA in the field):

[1] Raissi, M., Perdikaris, P. and Karniadakis, G.E., 2019. Physics-informed neural networks: A deep learning framework for solving forward and inverse problems involving nonlinear partial differential equations. Journal of Computational physics, 378, pp.686-707.
[2] Mohan, A.T., Lubbers, N., Livescu, D. and Chertkov, M., 2020. Embedding hard physical constraints in neural network coarse-graining of 3D turbulence. arXiv preprint arXiv:2002.00021.]

**Questions:**

Please see weaknesses

---

### Meta-Review · Area_Chair_nHYi · 2025-12-31

**Summary:**

This work studied the problem of denoising magnetometer data for magnetic anomaly navigation under GPS-denied environments. It integrates two physics-based constraints directly into neural network architectures to enhance model performance. The experimental results demonstrate the superior performance of the proposed method over the baselines.


Strength:

1. It addresses an important practical problem in navigation where GPS is unavailable or compromised. This research problem is interesting and practical.
2. It did some experiments to demonstrate the good performance of the proposed method.


Limitations:

1. The technical novelty is limited.

2. Do experiments on real-world data. It only evaluated the proposed method on synthetic data. It would be great to conduct experiments on real-world data.

3. The study needs to be compared against additional baseline methods. The lack of comparison with recent approaches makes the contribution difficult to assess.

4. The writing should be improved. The paper may require polishing and revision, particularly the figures.




In summary, this work requires significant refinement and improvement. It is not ready for publication.

**Reviewer Concerns:**

The authors did not respond to the comments

**Reviewer Scores:**

The authors did not respond to the comments from reviewers

---

### Decision · Program_Chairs · 2026-01-26

Reject